# Natural Enemies Acquire More Prey Aphids from Hormone-Treated Insect-Attracting Plants

**DOI:** 10.3390/plants14020147

**Published:** 2025-01-07

**Authors:** Xiaosheng Jiang, Xingrui Zhang, Guodong Han, Shovon Chandra Sarkar, Feng Ge

**Affiliations:** 1Shandong Key Laboratory for Green Prevention and Control of Agricultural Pests, Institute of Plant Protection, Shandong Academy of Agricultural Sciences, Jinan 250100, China; jiangxs98@163.com (X.J.); xy20126043@163.com (X.Z.); han_gd@foxmail.com (G.H.); 2College of Plant Protection, Nanjing Agricultural University, Nanjing 210095, China; 3College of Agriculture/Key Laboratory of Oasis Agricultural Pest Management and Plant Protection Resources Utilization, Xinjiang Uygur Autonomous Region, Shihezi University, Shihezi 832003, China; 4Food Futures Institute, Murdoch University, Murdoch, WA 6150, Australia; shovon47@gmail.com

**Keywords:** *Cnidium monnieri*, exogenous plant hormones, *Semiaphis heraclei*, *Orius minutus*, *Propylaea japonica*, life table

## Abstract

Exogenous plant hormones regulate the agronomic and physiological performance of plants and thus can influence the abundance of insect groups. We surveyed the arthropods on flowering plants *Cnidium monnieri* and found that the abundance of natural enemies *Propylaea japonica* and *Orius minutus* in the plots treated with salicylic acid (SA) and indole acetic acid (IAA) was significantly increased compared with those in the clean water (control) plots. Then, we investigated the effects of spraying SA, IAA, and clean water on the population parameters of *Semiaphis heraclei* reared on *C. monnieri.* Our results from the age-stage, two-sex life table analysis revealed a significantly shorter pre-adult duration for aphids reared on SA-treated *C. monnieri* compared to those reared on the other two treatments. The intrinsic rate of increase, finite rate of increase, and net reproductive rate of aphids reared on SA- and IAA-treated *C. monnieri* were significantly higher than those of aphids reared on clean water-treated *C. monnieri.* The fecundity rate was higher under the SA and IAA treatments than in the control, but the difference was not significant. This improved the ability of flowering plants to attract natural enemies by providing a larger food source.

## 1. Introduction

Plant hormones are chemical substances synthesized by plants that regulate their growth and development [1]. Exogenous plant hormones are artificial chemicals that have similar regulatory effects. Therefore, exogenously applied hormones can control the abundance of insect groups by regulating the agronomic and physiological performance of plants [2,3,4].

According to their physiological functions, plant growth regulators can be divided into plant growth promoters, retarders, and inhibitors. Indole acetic acid (IAA) is a plant growth promoter, while salicylic acid (SA) is a plant growth inhibitor [5]. IAA increases stomal conductance, photosynthetic rates, plant pigmentation, and the accumulation of total soluble sugars, such as fructose and glucose [6]. SA is also an exogenous plant hormone that can mediate crop resistance. However, experiments have found that this is not always the case. Because different concentrations of various hormones exhibit varied effects on different plants, it is important to verify these effects on test plants via experiments.

Hormones are central to plant physiology, metabolism, and development [7]. Exogenous hormones can regulate endogenous hormone signals [8], thus influencing the abundance of aphids on plants and, consequently, the presence of natural enemies. This occurs because natural enemies can spill over into the crop to consume aphids [9,10]. Therefore, it is important to study the indirect effects of plant growth regulators on insect populations.

Agroecological systems planted with *C. monnieri* can enhance ecological pest control [11,12,13]. *S. heraclei* serves as a food source for natural enemies, such as the ladybeetle species *Propylaea japonica* and other bug species like *Anthocoridae Orius minutus*. L. *S. heraclei* appears in early April, with its population peaking from mid to late May [14]. Moreover, our previous study showed that *C. monnieri* flowers can improve the longevity and reproductive ability of ladybeetles [15]. Therefore, *C. monnieri* is a favorable insect-attracting functional plant, and the number of *S. heraclei* on *C. monnieri* significantly influences its attraction ability [11,12]. Advancing the entry time of natural enemies is a promising strategy to enhance the ecological regulation ability of *C. monnieri.* As an insect-attracting functional plant, encouraging earlier blooming can help to attract natural enemies earlier.

Life tables provide information on the life history of an insect population [16]. The age-stage, two-sex life table theory is widely used since it involves both sexes, considers the variable developmental rates among individuals, and shows the stage differentiation of individuals in each population [11,16,17,18,19,20,21,22]. To date, no studies have focused on the indirect impact of exogenous plant hormones on the parameters of insects. Therefore, we used a life table to investigate the response of *S. heraclei* to the effects of exogenous plant hormones.

In this study, we used field surveys and life tables to explore how plant hormones enhance the ecological regulation ability of *C. monnieri* by affecting the abundance of aphids and natural enemies. We examined the incidence of life parameters influenced by plant hormones, explicitly the interaction between natural enemies and aphids under plant hormone treatment. The outcomes of this present study may serve as a framework for improving the ecological functions of functional plants.

## 2. Materials and Methods

### 2.1. Plant Exogenous Hormones

Salicylic acid (C_7_H_6_O_3_, lot. no.: 526M051, Shanghai yuanye Bio-Technology Co., Ltd., Shanghai, China, percentage composition ≥ 99.5%) and indoleacetic acid (C_10_H_9_NO_2_, lot. no.: J15GS151364, Beijing Solarbio Science & Technology Co., Ltd., Beijing, China, percentage composition ≥ 99%) were used in the experiments.

### 2.2. Insect Colony

A colony of *S. heraclei* was obtained from the Institute of Plant Protection, Shandong Academy of Agricultural Science, Jinan, Shandong, China. The colony was maintained on *C. monnieri* under greenhouse conditions at a temperature of 26 ± 1 °C, a relative humidity (RH) of 60 ± 10%, and an 8:14 h light–dark (L:D) photoperiod. New plants were transplanted into a greenhouse once a week to support the aphid population.

### 2.3. Field Survey

#### 2.3.1. Experimental Plot Design

A completely randomized block design was established in the experimental field plots. The field was divided into three blocks with nine plots overall. The total area of each plot was 81 m^2^ (3 m length × 3 m width). The three treatments were as follows: (1) sprayed with 150 mg/L SA on 26 April and 1 and 6 May in 2021 and on 15, 20, and 25 April in 2022; (2) sprayed with 150 mg/L IAA on 26 April and 1 and 6 May in 2021 and on 15, 20, and 25 April in 2022; and (3) sprayed with clean water on 26 April and 1 and 6 May in 2021 and on 15, 20, and 25 April in 2022. The sampling distances were defined as 5 m. The flower (*C. monnieri*) strips were sown on 9 September 2020 at a 37.5 kg seed/ha seeding density to ensure successful overwintering. The seeds were collected after maturation from plants previously grown by the laboratory in fields in Shandong Province. The seeds of the flower strips ripened in mid-July, fell to the ground, and germinated in 2020 and 2021. A 5 m isolation belt was left to minimize interference between plots, and a more than 5 m isolation belt was left between the experimental plot and its surroundings to minimize interference from the environment. No fertilizer, herbicides, or fungicides were used in the flower strips. All weeds in the flower strips were removed manually.

#### 2.3.2. *C. monnieri* Plot Cultivation

The plots were planted with *C. monnieri* in the fall of 2020 and 2021. The plants sprouted in winter and bloomed in the following spring. A solution of 150 mg/L SA and 150 mg/L IAA was sprayed on 26 April and 1 and 6 May in 2021 and on 15, 20, and 25 April in 2022. We sprayed clean water as the control treatment. We prepared a 10 L garden watering can and prepared the solution 1 h before each spray.

#### 2.3.3. Arthropod Sampling in the *C. monnieri* Field

The abundances of pests and enemies were all counted via visual observation within a 0.25 m^2^ (0.5 × 0.5 m) metal sampling frame. Five areas were selected within each plot for observation, and the average of these was calculated as the population size for the plot. Observations were carried out six times in 2021 and nine times in 2022, with a 7–10-day time interval.

### 2.4. Life Table Study

According to the results of previous field investigations, we choose a 150 mg/L SA treatment and an IAA treatment to study the influence of plant hormones on aphids.

For the control, A (150 mg/L SA), and B (150 mg/L IAA) treatments, the leaves were sprayed a total of three times at five-day intervals. Each treatment was conducted in a 10 × 30 m plot. The treated leaves (picked from the treated plants) were placed on the bottom of a 3.5 mm polystyrene acrylic Petri dish containing 5% agar, and then untreated mixed-age adult aphids were transferred onto the leaves. Ninety repetitions were prepared for each treatment. The dishes were stored in a growth chamber (27 ± 1 °C, RH = 70 ± 5%, and L:D = 12:12). After 24 h, the aphids were removed, two neonate nymphs were retained in each dish, and the results of the first day were recorded. After 24 h, one aphid was removed, only one aphid was retained, and each stage was recorded over time. The daily fecundity and death time were recorded every 12 h. The leaves and agar were replaced with fresh leaves and new agar every 3 days to ensure normal development of aphids.

### 2.5. Life Table Parameter Analysis

The life table data for *S. heraclei* were analyzed based on the age-stage, two-sex life table theory [16] and the method described by Chi [19] using the computer program TWOSEX-MSChart [23]. The age-stage-specific survival rate (*s_xj_*, where *x* = age and *j* = stage), age-specific survival rate (*l_x_*), age-stage-specific fecundity (*f_xj_*), and age-specific fecundity (m_x_), as well as the population parameters, including the net reproductive rate (*R_o_*), intrinsic rate of increase (*r*), finite rate of increase (*λ*), and mean generation time (*T*), were calculated according to the method described by Chi and Liu [23] and using the following equations, where *k* is the number of stages. The intrinsic rate of increase (*r*) was calculated using the Euler–Lotka formula with age indexed from day 0 [24]. The mean generation time was defined as the length of time it took a population to increase to *R*_0_-fold of its size in the stable age-stage distribution, and it was calculated as follows: *T* = (*ln R*_0_)/*r*. The age-stage life expectancy (*e_xj_*) for individuals of age x and stage j was calculated according to the method described by Chi and Su [25]. The age-stage reproductive value (*v_xj_*) is the contribution of an individual of age x and stage j to the future population [26].

### 2.6. Statistical Analysis

To assess the impact of various hormone treatments on the abundances of natural enemies and Semiaphis heraclei aphids, generalized linear mixed-effects models (GLMMs) were employed [27]. A zero-inflated Poisson model was used to analyze data on enemies, incorporating fixed effects for treatments (water, 150 mg/L salicylic acid, and 150 mg/L indoleacetic acid), year (2021, 2022), and their second-order interactions and a random effect for block. A zero-inflated negative binomial distribution was applied to aphid populations, incorporating fixed effects for different concentrations of SA and IAA and a random effect for blocks. The ‘emmeans’ package was utilized to carry out pairwise post hoc multiple comparisons of population numbers of natural enemies and aphids. The global model test for fixed effects in GLMM was run from the type II Wald chi-square tests via the ‘Anova’ function in the ‘car’ package. All the statistical analyses were conducted via R 4.2.0 software.

## 3. Results

### 3.1. Field Investigation Results

#### 3.1.1. Variation of Field Aphids Abundance Under Different Hormone Treatment

We investigated the natural enemies and aphid population dynamics in the field and analyzed the effect of plant hormones on them. We found that hormone treatment could delay the aphid population growth and decrease the abundance of aphids. The peak population of aphids in the clean water plot was observed on 29 May; however, hormone treatment, except the 50 mg/L SA treatment, advanced the peak population to between 2 May and 15 May, and the longest advance time was 14 days, which was observed in the 150 mg/L SA treatment plot (Figure 1 and Figure 2).

We observed that the aphid abundance decreased significantly under the hormone treatments (df = 4, *p* = 1.184 × 10^−8^), and the 150 mg/L SA treatment and IAA treatment were the most significantly decreased (150 mg/L SA treatment: *p* = 0.001; 150 mg/L IAA treatment: *p* = 0.004) (Figure 1).

#### 3.1.2. Variation in Field Natural Enemy Abundance Under Different Hormone Treatments

We investigated a variety of predators, including ladybeetles (Coleoptera: *Harmonia Axyridis*, *Coccinella septempunctata*, and *Adonia variegata*), hoverflies (*Diptera: Syrphidae*), and lacewings (*Neuroptera: Chrysopidae*), with no significant changes observed.

The results of the analysis showed that for ladybugs, there were significant differences between years for ladybeetles (Chisq = 29.1673, df = 1, *p* = 6.639 × 10^−8^) but no significant differences between treatments. However, *O. minutus* exhibited significant differences between years (Chisq = 15.4268, df = 2, *p* = 0.0004468) and treatments (Chisq = 48.3202, df = 1, *p* = 3.62 × 10^−12^). The results of pairwise comparisons also showed that treatment would significantly increase the population of *O. minutus* in 2022 (150 mg/L SA treatment: *p* = 0.0004 and 150 mg/L IAA: *p* = 0.0001) (Figure 2). Exogenous hormones increased at most of the survey time points, but the total number of natural enemies did not significantly increase in the whole growth period of the *Cnidium monnieri.* However, there was a certain degree of increase. During the two-year survey, the number of *O. minutus* was consistently higher in June than in May, while *Propylaea japonica* was more abundant in late May and mid-to-late June. This could be due to climatic conditions and the inherent characteristics of the species, leading to significant differences at only a few survey points. However, the number of individuals observed at the time was also slightly higher than in the control group, which could theoretically increase the level of biological control exerted by natural enemies.

### 3.2. Life Table Experiment

The parameters related to the developmental duration of each life stage, longevity, reproductive times, and fecundity of *S. heraclei* in the control and treatments (plant exogenous hormones) are shown in Table 1. The total duration of the pre-adult stage ranged from 5.46 d to 5.94 d, and the pre-adult development time was significantly longer in the control than in the treatments with SA and IAA. Adult longevity significantly increased with SA and IAA treatments; aphids on clean water-treated leaves had longer pre-oviposition periods than those on SA/IAA-treated leaves, with IAA-treated leaves exhibiting the longest APOP. The total pre-ovipositional period (TPOP), the duration from egg to first oviposition, of the aphids reared on leaves treated with SA and IAA was significantly shorter than that of the aphids reared on leaves treated with water. Additionally, the TPOP of the aphids reared on leaves treated with IAA was significantly longer than that of aphids reared on leaves treated with SA. There were no significant differences in fecundity between the control and the SA and IAA treatments.

For the age-stage-specific survival rate (sxj) (Figure 3), the curves show the survival rate, developmental stage, and overlap among the stages due to variable development rates among individuals. The survival rates of the aphids reared on leaves treated with SA (74%) and IAA (77%) were higher than those of the aphids reared on leaves treated with clean water (53%).

The age-specific survival rate (*l_x_*) and fecundity of *S. heraclei* are shown in Figure 4. lx is the sum of *s_xj_* at each stage x and is thus a simplified version of the *s_xj_* shown in Figure 3. The age-stage specific fecundity (*f_x_*) is the number of nymphs produced, and the age x is counted from the age-stage. The fecundity curve m_x_ shows that the periods from the initial time to reproduction in the control and two treatments ranged from 9 d (control and treatments) to 46 d (control), 47 d (IAA treatment), and 48 d (SA treatment). The maximum age-specific fecundity values were 2.98 (11 d), 3.12 (12 d), and 3.31 (12 d) for the nymphs in the SA, IAA, and control treatments, respectively.

The life expectancy (*e_xj_*) of *S. heraclei* at different ages and stages in the different treatments is plotted in Figure 5. Each plot in this curve represents the expected period for which an individual of age x and stage j will survive. The curves exhibit a decrease with age. The life expectancies of females were 16.1, 16.8, and 15.1 d for the SA, IAA, and control treatments, respectively.

The age-stage-specific reproductive values (*v_xj_*) of *S. heracleid* are plotted in Figure 6. The curve represents the contribution of an individual aphid of age x and stage j to the future population. The peak values of the age-stage reproductive value of females were 13.9 (SA treatment), 14.6 (IAA treatment), and 15.6 (control) at age 9, which indicates the highest contribution to the next generation.

The derived population parameters, including the intrinsic rate of increase, finite rate of increase, net reproductive rate, and mean generation rate, are presented in Table 2. The net reproductive rate (*R*_0_) of *S. heraclei* reared on leaves under the control treatment (*R*_0_ = 23.66) was significantly lower than those for the aphids reared on leaves under the SA (*R*_0_ = 37.67) and IAA (*R*_0_ = 37.70) treatments, with no significant difference observed between SA and IAA treatments. The highest intrinsic rate of increase (r) and finite rate of increase (*λ*) occurred in the SA treatment (*r* = 0.37; *λ* = 1.45) and were significantly higher than those of the IAA treatment. The lowest intrinsic rate of increase and finite rate of increase occurred in the control treatment (*r* = 0.28; *λ* = 1.33), with a significant difference observed between the IAA and the control treatments. The highest mean generation rate occurred in the control treatment (*T* = 11.22) and was significantly higher than that in the IAA treatment (*T* = 10.50). The lowest mean generation rate occurred in the SA treatment (*T* = 9.78), and there was a significant difference between the SA treatments and IAA treatments.

## 4. Discussion

IAA and SA treatments can significantly affect the field population of *S. heraclei* but significantly increase the laboratory population of *S. heraclei* by promoting nymph viability, adult fecundity, and lifecycle shortening. In addition, IAA can significantly increase *O. minutus* populations, while SA can increase *O. minutus* populations. Thus, we speculate that exogenous plant hormones influence the physical properties of plants and thus regulate pest insects on plants (e.g., aphids). We found that hormone treatment can advance the peak abundance date of aphids, delay aphid population growth, and finally, advance the peak abundance date of natural enemies. Natural enemies spill over into the crop to consume pest insects [9,10]. Thus, exogenous plant hormones may indirectly regulate the biocontrol ability of insect-attracting plants by advancing the peak abundance peak date of aphids, thereby providing an earlier food source for natural enemies and enabling them to breed and disperse sooner.

Only advancing the peak abundance time may not enhance the abundance of aphids and, therefore, provide additional food sources for natural enemies. In the present study, treated plants increased the fecundity, duration of the adult period, and total life span of the *S. heraclei* population. Exogenous plant hormones regulate agronomic and physiological performance, further influencing the endogenous hormones and sugar content in plants, which, in turn, affects the abundance of insect populations [3,4,28]. The indirect impact of exogenous plant hormones on a population of insects significantly increased the net reproductive rate (*R*_0_), intrinsic rate of increase (*r*), finite rate of increase (*λ*), and mean generation time (*T*) of *S. heraclei.* This means that treating *C. monnieri* with IAA and SA can provide a better habitat and nutrition for *S. heraclei.*

Few studies have documented that exogenous plant hormones can indirectly affect insect populations on plants [20]. The application of methyl jasmonate can have positive effects on attracting natural enemies and promoting a pest control function in cotton fields [29]. The external application of gibberellin and jasmonic acid has positive effects on the natural enemy attraction (indirect defense) and self-insect resistance (direct defense) of rice [26]. The application of methyl jasmonate and salicylic acid can inhibit the growth of the citrus woodlouse and attract natural enemies. SA is a plant hormone that can mediate several types of plant physiological processes [30,31]. IAA can regulate body weight, trehalose content, detoxification activity, and metabolic enzymes in aphids [32]. IAA can also regulate the induction of galls [32]. In addition, the exogenous application of SA can be used to reduce settling behavior and attract natural enemies by promoting the volatile content of citrus foliage [33]. In addition, hormones can significantly change the physical properties of plants [34]. This may change the microenvironment of insect habitats, thus significantly influencing the abundance and distribution of insects [35]. In this study, the duration of the first and second stages of *S. heraclei* was significantly decreased. This may have increased its adaptation, and the fecundity of *S. heraclei* was also increased in the SA and IAA treatments. The results of the laboratory experiments are contrary to the results of the field survey results. This may be due to the occurrence of predation by natural enemies in the field. Because *C. monnieri* is part of the *Apiaceae* family and plant growth regulators can regulate its physiological characteristics, these changes may facilitate predators in preying on aphids while also providing a better habitat for the predators.

The insect-attracting plant *C. monnieri* can provide a habitat and food source for the natural enemies of aphids [27,36], and we have found that adding the flower *C. monnieri* can more significantly promote *Harmonia axyridis* viability than water and other foods [15]. Previous studies have mostly improved different types of flowering plants through habitat management to attract more natural enemies and enhance the prevention effect [37,38,39]. After two years of investigation, we found that hormones could increase the abundance of *P. japonica* and *O. minutus*. Exogenous plant hormones change the physical structure of plants, which may change their microenvironment and, thus, influence insect colonization. Microenvironment parameters include nutrition, luminosity, temperature, and relative humidity and depend on the physical structure of plants [40,41]. These parameters can determine the adaptation of insects to plants [2,35,42,43,44,45,46,47]. The SA and IAA treatments directly changed the physical structure of *C. monnieri* and indirectly regulated the aphid populations and advanced peak abundance time. Pollen, nectar, and alternative prey may also influence these results, but this requires further study. In this study, the population parameters of aphids in Petri dishes revealed that treating *C. monnieri* with SA and IAA could decrease the lifespan and enhance the fecundity of *S. heraclei*, thereby increasing its population growth. This means that *C. monnieri* can provide a more abundant food source for the natural enemies of *S. heraclei*. Further cage experiments are necessary to confirm this link more robustly.

## 5. Conclusions

Treating *C. monnieri* with SA and IAA enhances the survival, longevity, and reproductive capabilities of *S. heraclei* while reducing its developmental period. As a result, these treatments allow the plant to provide a more abundant and earlier food source, attracting and sustaining a greater number of natural enemies. This innovative approach utilizes a single flowering plant to attract diverse natural predators, effectively regulating aphid populations on wheat. Consequently, it enhances the ecological regulation capacity of *C. monnieri*, demonstrating a promising strategy for pest control.

## Figures and Tables

**Figure 1 plants-14-00147-f001:**
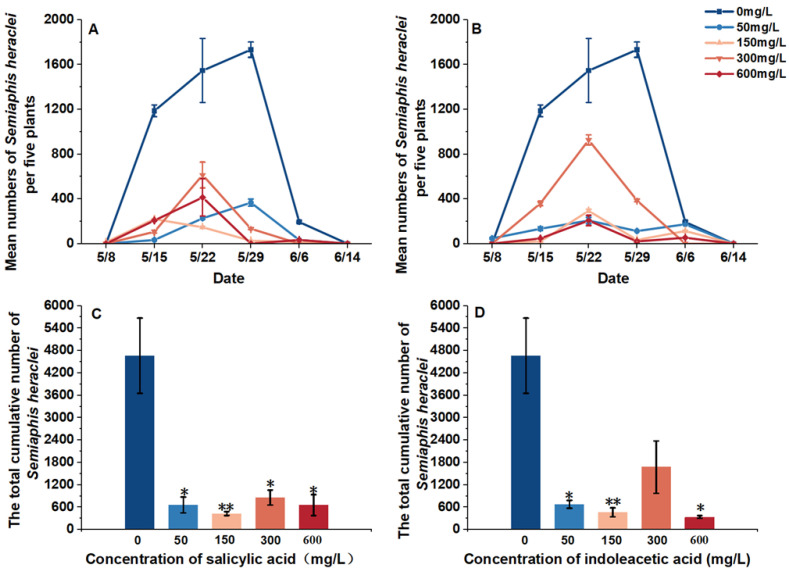
Response of aphid populations to different treatments. (**A**) Population dynamics of aphids *Semiaphis heraclei* on 5 flowering plants of *Cnidium monnieri*, which were treated with salicylic acid; (**B**) Population dynamics of *Semiaphis heraclei* on 5 flowering plants of *Cnidium monnieri*, which were treated with indoleacetic acid; (**C**) Total cumulative number of *Semiaphis heraclei* during all the sampling periods (5 times) according to the visual observations of the salicylic acid-treated *Cnidium monnieri*; (**D**) Total cumulative number of *Semiaphis heraclei* during all the sampling periods (5 times) according to the visual observations of the indoleacetic acid-treated *Cnidium monnieri* (due to the climate conditions in 2022, the number of aphids on the entire plot was very low that year, so data from 2022 were not included). * means *p* ≤ 0.05; ** means *p* ≤ 0.01.

**Figure 2 plants-14-00147-f002:**
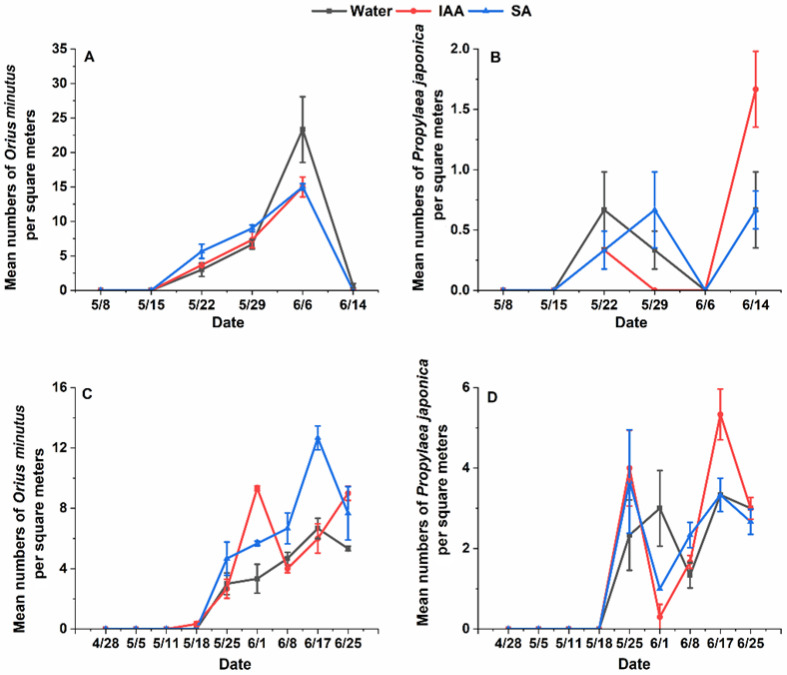
Population dynamics of natural enemies on flowering plants treated with water 150 mg/L, salicylic acid, and 150 mg/L indoleacetic acid in 2021 and 2022. (**A**): Population dynamics of *Orius minutus* on one square meter of *Cnidium monnieri* in 2021; (**B**): Population dynamics of *Propylaea japonica* on one square meter of *Cnidium monnieri* n 2021; (**C**): Population dynamics of *Orius minutus* on one square meter of *Cnidium monnieri* n 2022; (**D**): Population dynamics of *Propylaea japonica* on one square meter of *Cnidium monnieri* in 2022.

**Figure 3 plants-14-00147-f003:**
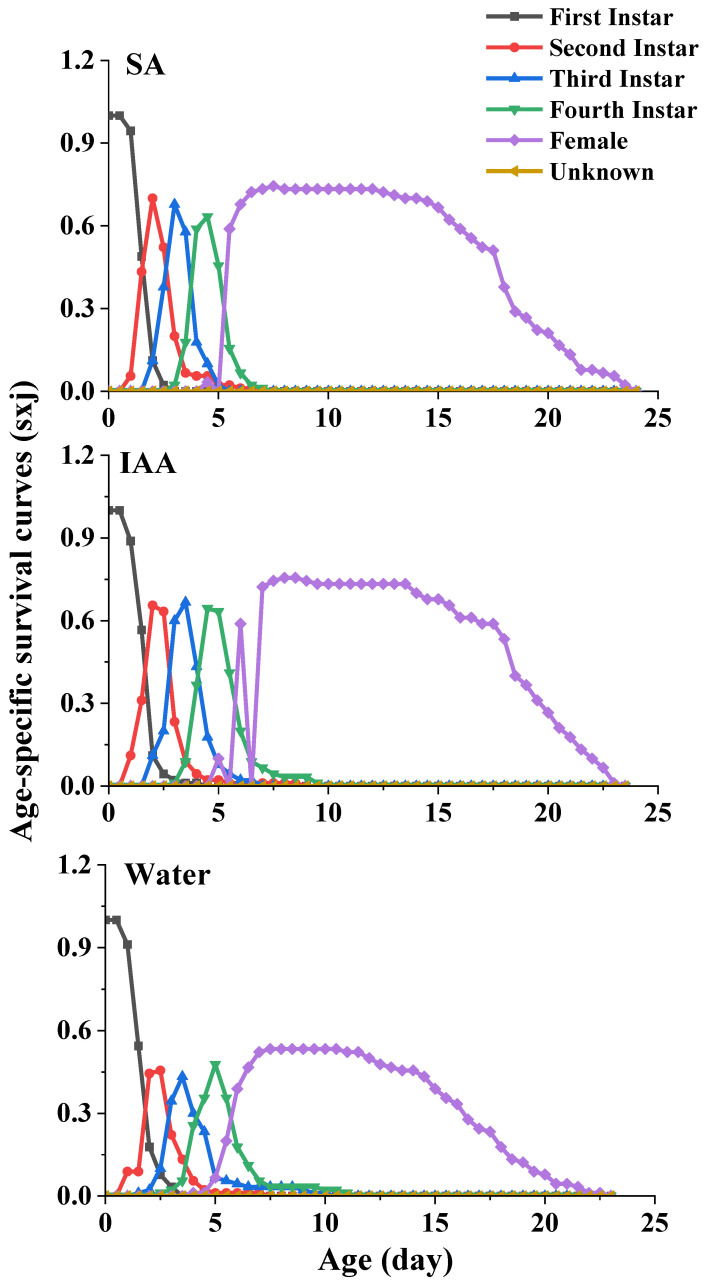
Age-specific survival curves (Sxj) of *Semiaphis heraclei* treated with salicylic acid and indoleacetic acid.

**Figure 4 plants-14-00147-f004:**
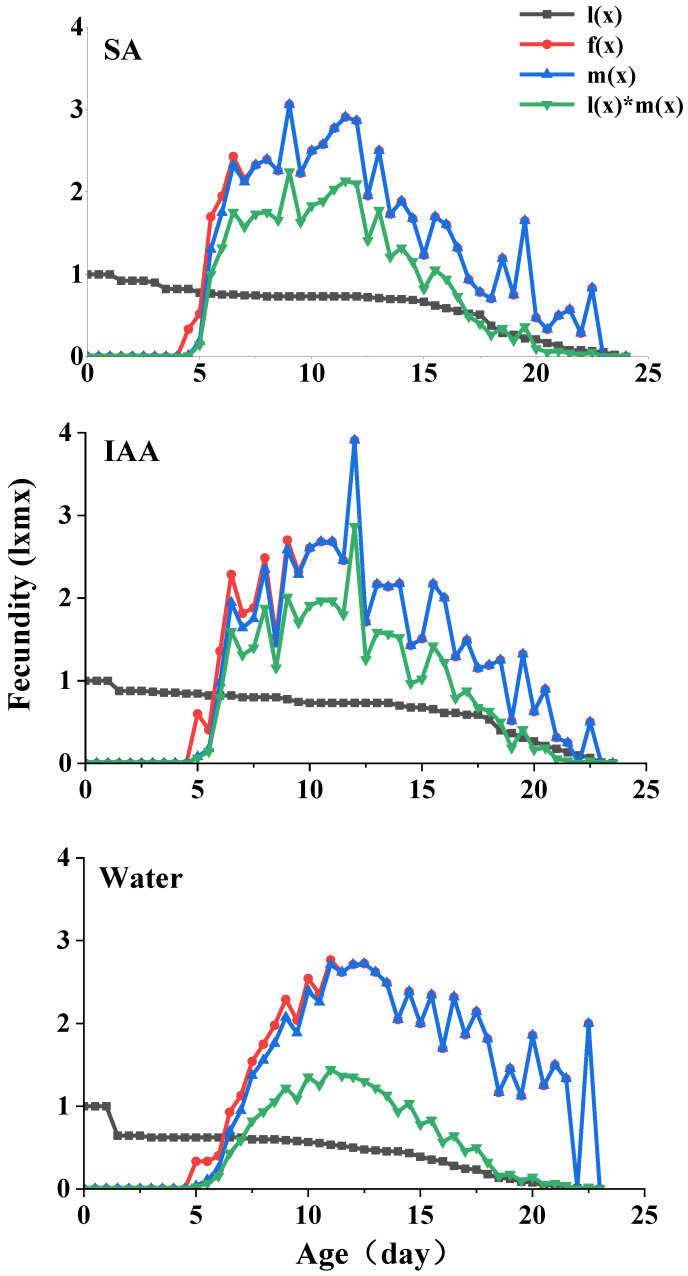
Age-specific survival rate (lx), age-specific reproduction rate (mx), age-specific reproduction rate (fx), and age-specific reproduction value (lxmx) in parthenogenetic populations of *Semiaphis heraclei* treated with salicylic acid and indoleacetic acid.

**Figure 5 plants-14-00147-f005:**
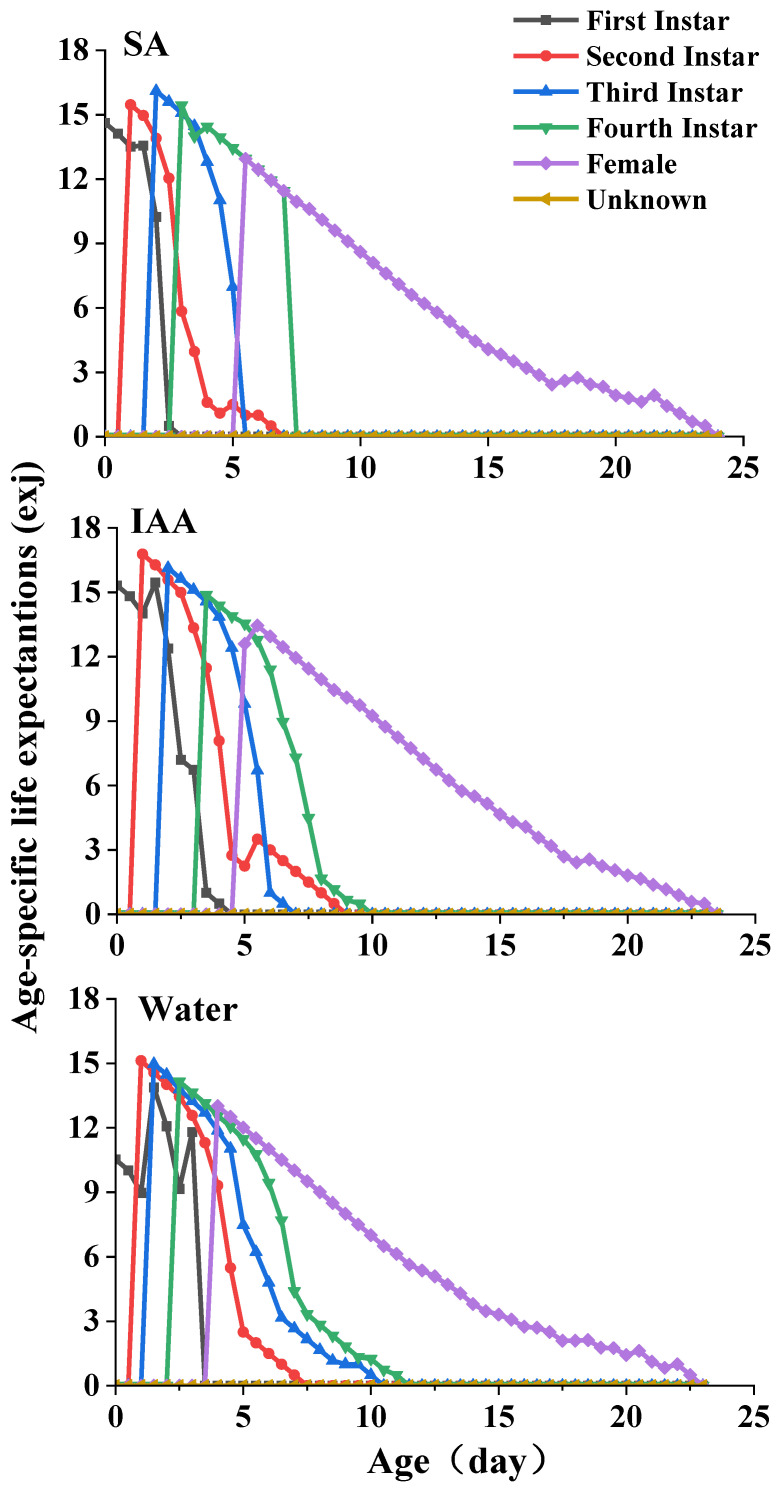
Age-specific life expectation of *Semiaphis heraclei* treated with salicylic acid and indoleacetic acid exj.

**Figure 6 plants-14-00147-f006:**
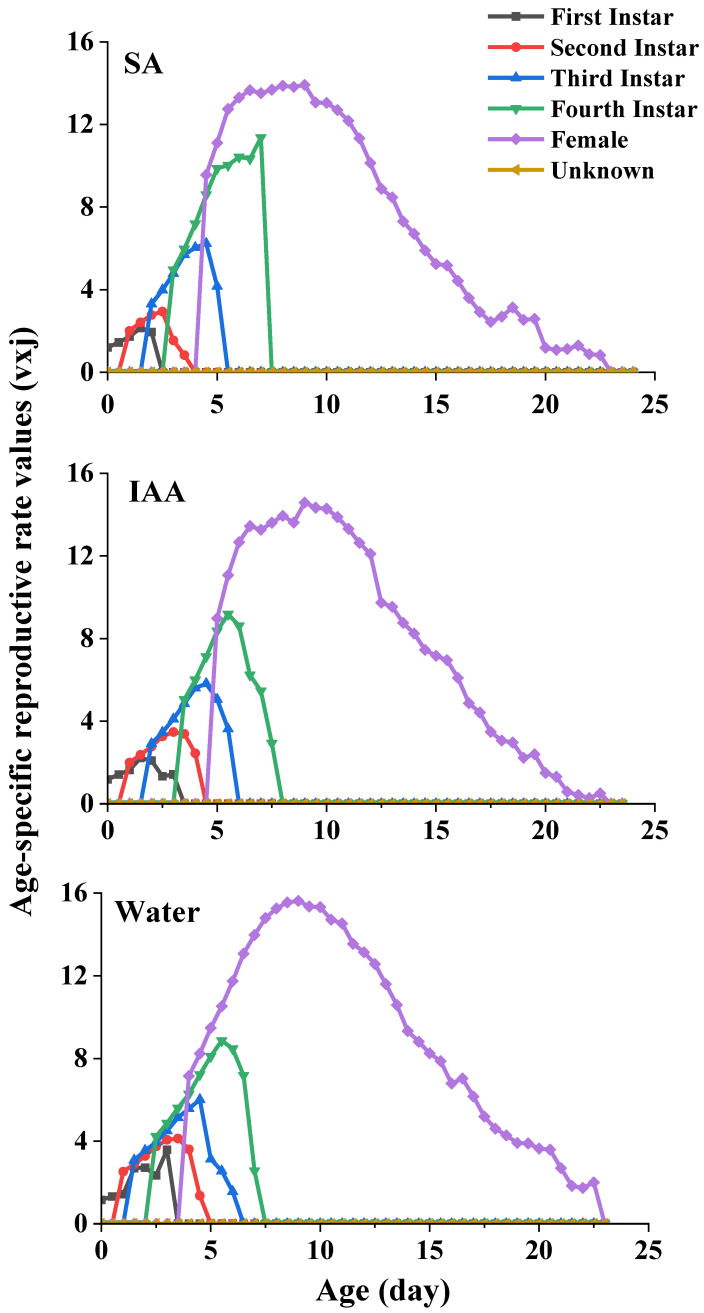
Age-specific reproductive rate values of *Semiaphis heraclei* treated with salicylic acid and indoleacetic acid (vxj).

**Table 1 plants-14-00147-t001:** Developmental duration, longevity, and mean fecundity of *Semiaphis heraclei* under the control, SA, and IAA treatments.

Parameters		Control		Treatments		
	n	Clean Water	n	SA	n	IAA
First stage	56	2.04 ± 0.08 a	81	1.78 ± 0.042 b	77	1.82 ± 0.054 ab
Second stage	55	1.21 ± 0.05 a	69	0.96 ± 0.047 b	74	1.2 ± 0.048 a
Third stage	52	1.28 ± 0.06 a	67	1.28 ± 0.052 a	72	1.36 ± 0.036 a
Fourth stage	48	1.49 ± 0.06 a	67	1.43 ± 0.041 a	69	1.51 ± 0.043 a
Adult	48	11.07 ± 0.39 b	67	12.99 ± 0.362 a	69	12.89 ± 0.41 ab
Pre-adult	48	5.94 ± 0.10 a	67	5.46 ± 0.068 b	69	5.86 ± 0.076 a
Total	90	10.52 ± 0.80 b	90	14.62 ± 0.748 a	90	15.32 ± 0.747 a
Fecundity (F)	72	44.35 ± 0.87 a	80	50.60 ± 1.057 a	78	49.17 ± 1.203 a
APOP	48	1.41 ± 0.080 a	67	0.35 ± 0.046 b	69	0.54 ± 0.058 c
TPOP	48	7.34 ± 0.16 a	67	5.81 ± 8.082 b	69	6.4 ± 0.086 c

The table shows significant differences, a, b and c represent significant differences between groups, *p* < 0.05.

**Table 2 plants-14-00147-t002:** Population parameters (mean ± SE) of *Semiaphis heraclei* in control and treatment.

Parameters	SA	IAA	Clean Water
Intrinsic rate of increase, r (d^−1^)	0.37 ± 0.003 a	0.35 ± 0.001 b	0.29 ± 0.003 c
Finite rate of increase, λ (d^−1^)	1.45 ± 0.004 a	1.41 ± 0.004 b	1.33 ± 0.003 c
Net reproductive rate, R_0_	37.67 ± 0.574 a	37.70 ± 1.045 a	23.66 ± 0.629 b
Mean generation rate T, (d)	9.78 ± 0.054 c	10.50 ± 0.026 b	11.22 ± 0.021 a

The table shows significant differences, a, b and c represent significant differences between groups, *p* < 0.05.

## Data Availability

Data is contained within the article.

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
