# Peer review of "Natural Enemies Acquire More Prey Aphids from Hormone-Treated Insect-Attracting Plants"

_plants, 2025, doi:10.3390/plants14020147_

Round 1
Reviewer 1 Report
Comments and Suggestions for Authors
"The reviewer generally agrees that 'IAA and SA treatment can significantly affect the field population of S. heraclei, and increase the laboratory population by promoting nymph viability, adult fecundity, and shortening the life cycle.' However, the results showing the connection between IAA/SA treatment and the population of natural enemies were unclear and require significant revision to make them clearer for readers. Additionally, many sentences have issues with clarity, and there are spelling and grammatical errors throughout the manuscript that need to be corrected."
l In various parts of the manuscript, the authors refer to Orius minutus, but the figures only show results for Orius similis. This discrepancy needs to be corrected throughout the manuscript for consistency. Please ensure that the species mentioned in the text matches the one presented in the figures.
Orius similis: Found primarily in parts of East Asia, such as China, Korea, and Japan. It is commonly used in biological control in greenhouse and agricultural settings for its effectiveness against thrips.
Orius minutus: More common in Europe and parts of Asia. It also preys on small pests, but its distribution and ecological niche differ slightly from O. similis.
l Line 155-159, “We found that hormone treatment can advance the peak date of aphids and decrease the abundance of aphids. The peak population of aphids in CK plot was at 5.29,but hormone treatment except 50mg/L SA treatment can make the peak advance to 5.22 even to 5.15.The longest advance time was 14 days observed in 150mg/L SA treatment plot (Figure 1 and 2).” à According to Fig. 1, the peak population of aphids in CK plot was not significantly different in both 5/29 and 5/22. Moreover, no data is shown for peak at 5/15
l Line 181-184, “In 2021 the number of O. minutus in SA treatment plot and IAA treatment plot were larger than water treatment plot on 5.22 and 5.29, but lower on 6.6 with no significant. The density of P. japonica significantly increased in IAA treatment (F = 1.272, df = 27, p = 0.03) (Fig. 3).” à "In Fig. 3, the results are presented for the aphid Semiaphis heraclei, not Orius minutus as stated in the text. Additionally, Fig. 2 shows results for Orius similis, not Orius minutus.
l Line 185-191 “In 2022, the number of O. minutusa in SA treatment plot and IAA treatment plot were bulk larger than water treatment plot and the number of P. japonica were n SA treatment plot and IAA treatment plot were larger than water treatment plot except 6.1. The number of P. japonica treated with IAA significantly increased on May 25 (F = 3.588, df = 18, p = 188 0.005) (Fig. 3). The number of O. minutus in 2022 in treatment IAA significantly increased (F = 0.976, df = 18, p = 0.047), and in SA treatment on June 8 (F = 5.897, df = 18, p = 0.007) (Fig. 2).” à The significant results are only observed at specific time points, rather than showing consistent patterns across the entire study period. It would be helpful for the authors to clarify whether these findings represent isolated events or if there are underlying factors contributing to the variation over time. Further discussion or analysis on the temporal dynamics would strengthen the interpretation of the results.
Line 52 attrcat à attract
Line 57 rovide à provide
Line 69 C. monnieri.C. monnieri à C. monnieri. C. Monnieri
Line 80 by affect à by affecting
Line 154-155 analysis the affection of plant hormone à analyze the effect by plant hormones
Line 157 CK plot? 5.29à 5/29
Line 158 even to 5.15.The à no 5/15 and The
Line 161-163 (F = 1.131, df = 27, p = 0.007 for 150mg/L SA treatment; F = 1.131, df = 27, p = 0.008 for 150mg/L IAA treatment) F? df? P? no methods for calculation à F=ANOVA coefficient
Fig. 2 y-axis pear to per
Line 186 were n à maybe “in”
Comments on the Quality of English Language
Too much typing errors and unclear presentation
Author Response
Comments 1:In various parts of the manuscript, the authors refer to Orius minutus, but the figures only show results for Orius similis. This discrepancy needs to be corrected throughout the manuscript for consistency. Please ensure that the species mentioned in the text matches the one presented in the figures.
Orius similis: Found primarily in parts of East Asia, such as China, Korea, and Japan. It is commonly used in biological control in greenhouse and agricultural settings for its effectiveness against thrips.
Orius minutus: More common in Europe and parts of Asia. It also preys on small pests, but its distribution and ecological niche differ slightly from O. similis.
Reply:I alter Orius similis has been change to Orius minutus . Because this study was in ShanDong,and Orius minutus distribute in ShanDong.
Comments 2:Line 155-159, “We found that hormone treatment can advance the peak date of aphids and decrease the abundance of aphids. The peak population of aphids in CK plot was at 5.29,but hormone treatment except 50mg/L SA treatment can make the peak advance to 5.22 even to 5.15.The longest advance time was 14 days observed in 150mg/L SA treatment plot (Figure 1 and 2).” à According to Fig. 1, the peak population of aphids in CK plot was not significantly different in both 5/29 and 5/22. Moreover, no data is shown for peak at 5/15
Reply. We pay more attention to whether the peak period is earlier, and pay less attention to the significance of the difference in quantity. However, it can also be seen that the quantity of blue ck increases to a certain extent compared with that of exogenous hormone treatment. The yellow line of 150mg/L in Figure 1A, which is advanced to 15th, may not be visible to you because it is not obvious and overlaps with other lines.
Comments 1:Line 181-184, “In 2021 the number of O. minutus in SA treatment plot and IAA treatment plot were larger than water treatment plot on 5.22 and 5.29, but lower on 6.6 with no significant. The density of P. japonica significantly increased in IAA treatment (F = 1.272, df = 27, p = 0.03) (Fig. 3).” à "In Fig. 3, the results are presented for the aphid Semiaphis heraclei, not Orius minutus as stated in the text. Additionally, Fig. 2 shows results for Orius similis, not Orius minutus.
Reply:Orius similis has been change to Orius minutus .
Comments 3:Line 185-191 “In 2022, the number of O. minutusa in SA treatment plot and IAA treatment plot were bulk larger than water treatment plot and the number of P. japonica were n SA treatment plot and IAA treatment plot were larger than water treatment plot except 6.1. The number of P. japonica treated with IAA significantly increased on May 25 (F = 3.588, df = 18, p = 188 0.005) (Fig. 3). The number of O. minutus in 2022 in treatment IAA significantly increased (F = 0.976, df = 18, p = 0.047), and in SA treatment on June 8 (F = 5.897, df = 18, p = 0.007) (Fig. 2).” à The significant results are only observed at specific time points, rather than showing consistent patterns across the entire study period. It would be helpful for the authors to clarify whether these findings represent isolated events or if there are underlying factors contributing to the variation over time. Further discussion or analysis on the temporal dynamics would strengthen the interpretation of the results.
Reply:During the two-year survey, the number of O. minutus was consistently higher in June than in May, while Propylaea japonica were more abundant in late May and mid-to-late June. This could be due to climatic conditions and the inherent characteristics of the species, leading to significant differences at only a few survey points. However, the number of individuals observed at the time was also slightly higher than in the control group, so theoretically, could theoretically increase the level of harm control of natural enemies.
Comments 4:Line 52 attrcat à attract
Reply:attrcat has been change to attract
Comments 5:Line 57 rovide à provide
Reply:rovide has been change to provide
Comments 6:Line 69 C. monnieri.C. monnieri à C. monnieri. C. Monnieri
Reply:C. monnieri.C. monnieri has been change to C. monnieri. C. Monnieri
Comments 7:Line 80 by affect à by affecting
Reply:by affect has been change to by affecting
Comments 8:Line 154-155 analysis the affection of plant hormone à analyze the effect by plant hormones
Reply: analysis the affection of plant hormone has been change to analyze the effect by plant hormones
Comments 9:Line 157 CK plot? 5.29 à 5/29
Reply: CK plot? 5.29 has been change to 5/29
Comments 10:Line 158 even to 5.15.The à no 5/15 and The
Reply: even to 5.15.The has been change to no 5/15 and The
Comments 11:Line 161-163 (F = 1.131, df = 27, p = 0.007 for 150mg/L SA treatment; F = 1.131, df = 27, p = 0.008 for 150mg/L IAA treatment) F? df? P? no methods for calculation à F=ANOVA coefficient
Reply: These values are calculated in the SPSS software, hence the calculation method is not detailed here.
Comments 12:y-axis pear to per
Reply: I don’t understand this,so not modified for now
Comments 13:Line 186 were n à maybe “in”
Reply: were n has been change to maybe “in”

Reviewer 2 Report
Comments and Suggestions for Authors
Dear authors,
Here are my comments about your work.
INTRODUCTION:
-Line 52: change attrcat by attract.
-Line 53: after a point, write scientific names without abbreviation. In this line and others (Line 60, 69).
-Line 57: change rovide by provide.
-General comment: you focused your study on aphids as a source of food for natural enemies. To see that you study life tables only for aphids. I think you should study also natural enemies life tables. You do not know if hormones in C. monnieri have a positive effect on both, aphids and natural enemies.
MATERIALS AND METHODS:
-Line 87: add information about the origin of chemicals (company). I know because of results that you check several hormones concentrations. You should put this information here.
-Line 100: how many plots per block?
-Line 103 and 106: scientific names in intalics.
-Line 104: information about each spray and concentration here.
-Line 108: how many plots per block?
-Line117: this is the third plot size that I read. Explain better this information. Did you use the same plots for field and life table studies?
-Line 147: an ANOVA is not the best way to analyze time series. I suggest a GLMM with random factors as blocks.
RESULTS:
-Line 153 and 180: change vary by variation.
-Line 157: what is the meaning of CK? Change dates format 5.29 looks a number.
-Line 159: remove fig 2 from here.
-Line 184: remove fig 3 from here.
-Table 1: treatment in plural.
-Line 216: is this significantly different?
DISCUSSION:
-Line 328: did you found other natural enemies?
General comment: you focus only in C. monnieri, but you did not check natural enemies on the crop, which is very important.
Regards!
Author Response
Comments 1:Line 52: change attrcat by attract.
Reply:attrcat has been change to attract.
Comments 2:Line 53: after a point, write scientific names without abbreviation. In this line and others (Line 60, 69).
Reply:Has change this to right format
Comments 3:Line 57: change rovide by provide.
Reply: rovide has been changed to provide.
Comments 4:General comment: you focused your study on aphids as a source of food for natural enemies. To see that you study life tables only for aphids. I think you should study also natural enemies life tables. You do not know if hormones in C. monnieri have a positive effect on both, aphids and natural enemies.
Reply:I really appreciate your input, which provides significant additions to the content and meaning of the article. However, due to the limitations of experimental conditions and the fact that I have left the original laboratory, it is difficult to add further. Once again, I am grateful for your suggestions.
Comments 5:Line 87: add information about the origin of chemicals (company). I know because of results that you check several hormones concentrations. You should put this information here.
Reply:I have add the company information.
Comments 6:Line 100: how many plots per block?
Reply:Three plots per block,I add this.
Comments 7:Line 103 and 106: scientific names in intalics.
Reply:I have changed this
Comments 8:Line 104: information about each spray and concentration here.
Reply:150 mg/L SA and 150 mg/L IAA
Comments 9:Line 108: how many plots per block?
Reply:Five plots per block,I add this.
Comments 10:Line117: this is the third plot size that I read. Explain better this information. Did you use the same plots for field and life table studies?
Reply:I conducted three replicates for each hormone treatment, and during the survey, I selected five sample points in each replicate plot to investigate the number of insects.
Comments 11:Line 147: an ANOVA is not the best way to analyze time series. I suggest a GLMM with random factors as blocks.
Reply:GLM is a very good analytical method, but many studies in journals show that ANOVA is also a viable method. Is it possible to continue using it?
Comments 12:Line 153 and 180: change vary by variation.
Reply:vary has been changed to variation.
Comments 13:Line 157: what is the meaning of CK? Change dates format 5.29 looks a number.
Reply:5.29 has been changed to May 29,other same problems were also change.CK is the treatment of water, I have add this information.
Comments 14:Line 159: remove fig 2 from here.
Reply: I have changed this.
Comments 15:Line 184: remove fig 3 from here.
Reply: I have changed this.
Comments 16:Table 1: treatment in plural.
Reply:treatment has been changed to treatments
Comments 17:Line 216: is this significantly different?
Reply: This is significantly different:
Comments 18:Line 328: did you found other natural enemies?
Reply: I surveyed many natural enemies, including important ones such as the exotic ladybug, the tortoise ladybug, aphid flies, and green lacewings. However, since there were no significant differences in the analysis, they were not included in this article.
Comments 19:General comment: you focus only in C. monnieri, but you did not check natural enemies on the crop, which is very important.
Reply:I think your point is very correct. We had also considered this situation at the time, but due to the division of work, it was not possible to grow crops in my experimental field. Subsequently, other members of the group will conduct a more in-depth study on your suggestions.

Reviewer 3 Report
Comments and Suggestions for Authors
this study aims to describe the indirect impact of vegetal hormone on aphids and their predators.
M&M : give more details on the methods used to calculate each parameters. You dont give the number of blocks in your experiment.
"the mean number of predatory natural enemies per square meter and aphids per 5 plants were assessed at different sampling 109 dates on C. monnieri."
113 : "previous" field investigation
l114 : i dont understand "lucubrate"
l122 : why do you remove one of the two aphid after 24h ? or why keep two initialy ?
l117 : why treating a 10x30m plot when you just explained before the plots were 3x3m in 2.3.1. or give a experimental plan ?
l140 are you sure the reference is appropriate ?
Results
l156 : I dont understand how you can deduce this result from your graphs. I can indeed see that aphid population is reduced when treated , but considering the peack date this is quite tricky : the curve trajectory is way higher and start the increasing way sooner.
figure 1 et cie, remove the "The" in front of the concentration rates in the absiss axis
l282 : i dont understand epeculate (is it speculate ?)
l284 "abundance" instead of "adundance"
l306 . missing
and so on, please revise
l320 : you have the number of preys and predators, you should test the hypothesis you suggest, to see wether you would have a link between numbers of predators present and the impact on preys : by the look of your figures 1 and 2 i am really not sure ! the numbers of orius in both years was quite equivalent between treatments so...
I think you have to go deeper to explain why you have such a difference between lab and field experiment. this is a very tricky point.
Comments on the Quality of English Language
the manuscript english is not very comprehensible, a big language revision is required. an effort is also required on the figures legends.
blanks missing after points
Author Response
Comments 1:M&M : give more details on the methods used to calculate each parameters. You dont give the number of blocks in your experiment.
Reply:These values are calculated in the SPSS software, hence the calculation method is not detailed here.These values are calculated in the SPSS software, hence the calculation method is not detailed here.
Comments 2:113 : "previous" field investigation
Reply: I have add previous.
Comments 3:114 : i dont understand "lucubrate"
Reply:lucubrate has been changed to study
Comments 4:122 : why do you remove one of the two aphid after 24h ? or why keep two initialy ?
Reply:The method is primarily acquired through literature. Placing two aphid is to prevent the death of fragile first-instar nymphs, which could lead to insufficient repetition in handling. Removing one aphid is done once the aphids have successfully colonized the leaf, to ensure their normal development and to prevent unnecessary effects due to overcrowding of aphids at both ends.
Comments 5:117 : why treating a 10x30m plot when you just explained before the plots were 3x3m in 2.3.1. or give a experimental plan ?
Reply:This is a specific plot designated for providing experimental leaves for life tables, separate from the aforementioned population survey plot. Choosing a different plot is to avoid affecting the population survey. The size of 10×30 is intended to provide a sufficient number of leaves.
Comments 6:140 are you sure the reference is appropriate ?
Reply:I am sure.
Comments 7:156 : I dont understand how you can deduce this result from your graphs. I can indeed see that aphid population is reduced when treated , but considering the peack date this is quite tricky : the curve trajectory is way higher and start the increasing way sooner.
Reply:We pay more attention to whether the peak period is earlier, and pay less attention to the significance of the difference in quantity. The high number of CK treatments is speculated to be due to fewer natural enemies on the plants. The peak in aphid numbers after hormone treatment indeed occurs earlier.
Comments 8:figure 1 et cie, remove the "The" in front of the concentration rates in the absiss axis
Reply:i had remove "The"
Comments 9:282 : i dont understand epeculate (is it speculate ?)
Reply: epeculate has been changed to speculate
Comments 10:284 "abundance" instead of "adundance"
Reply:adundance has been changed to abundance
Comments 11:306 . missing and so on, please revise
Reply:I have revise all
Comments 12:320 : you have the number of preys and predators, you should test the hypothesis you suggest, to see wether you would have a link between numbers of predators present and the impact on preys : by the look of your figures 1 and 2 i am really not sure ! the numbers of orius in both years was quite equivalent between treatments so...
I think you have to go deeper to explain why you have such a difference between lab and field experiment. this is a very tricky point.
Reply:You've given me a very useful suggestion, and I will conduct further analysis on this in subsequent articles.
Comments 13:the manuscript english is not very comprehensible, a big language revision is required. an effort is also required on the figures legends.
Reply:I will make further revisions to this, and after there are no other issues, I will polish the entire thesis.

Reviewer 4 Report
Comments and Suggestions for Authors
The submitted manuscript deals with an important issue of exogenous application of plant hormones to enhance the plant attractiveness, namely Cnidium monnieri, to the aphid species Semiaphis heraclei and, in consequence, promote the abundance of aphid natural enemies, the ladybird Propylaea japonica and the bug Orius similis.
To investigate this issue the Authors treated the plants with salicylic acid and indoleacetic acid, then calculated the life parameters of the aphid and the abundance of natural enemies.
The methods used for the study were correct and the data collection during two consecutive vegetation seasons should provide enough data for interpretation.
Unfortunately, the manuscript is written in a very unclear style and various issues need clarification.
The first issue is the problem with definitions of endogenous and exogenous plant hormones. Your statement that ‘exogenous plant hormones are artificial chemicals with similar regulatory effects’ (line 31-32) is completely untrue. ‘Exogenous’ in your experiments means ‘exogenously applied’. You use natural plant hormones for your experiment and not some artificial compounds that have different molecules than IAA or SA. Both IAA and SA are endogenous natural plant hormones that regulate various natural life processes of plants. In addition, SA has an important role in plant signaling in response to pathogens and aphids. You should provide more correct definitions in the Introduction.
The second issue is the lack of information about the plant and the aphid species. The systematic position of the plant should be provided (e.g., botanical family, occurrence, ecological roles, volatiles attracting entomophages – what volatiles? – line 52). In the case of the aphid, more information would not hurt: for example, a short description of the life cycle, the host plant range to demonstrate that it is not a threat to cultivated plants, etc. The information on the aphid is necessary for the understanding of the results of the experiment: I assume that you studied the parthenogenetic population of the aphid. At this point in the life history, I assume that aphids reproduced parthenogenetically and viviparously. Therefore, the statement ‘eggs produced’ – line 220 – what eggs? Aphididae do not lay eggs when reproduce in the vegetative season on the secondary hosts. Literature provides the information that S. heraclei is a holocyclic host alternating species. In summer, it should be viviparous. Is it not? Please, clarify.
What is ‘preadult stage’ in the aphid (Table 1)? Do you mean the winged nymph? How many larval stages has this aphid? You can see, that the information on the aphid would be very useful here.
Table 1. Which data refer to what? Where is longevity?
Figure 1. What year of study does this figure refer to? Why only one year is presented while Figure 2 presents the two-year results.
I listed only the most important issues crucial for the understanding of the results. As there are so many doubts, I can not evaluate the correctness of the interpretation. The article must be substantially improved. I am sure you have all necessary data but more work should be done. Moreover, the English of the text needs a lot of editing.
Comments on the Quality of English LanguageA lot of grammatical errors and wrong uses of words.
Author Response
Round4
Comments 1:The first issue is the problem with definitions of endogenous and exogenous plant hormones. Your statement that ‘exogenous plant hormones are artificial chemicals with similar regulatory effects’ (line 31-32) is completely untrue. ‘Exogenous’ in your experiments means ‘exogenously applied’. You use natural plant hormones for your experiment and not some artificial compounds that have different molecules than IAA or SA. Both IAA and SA are endogenous natural plant hormones that regulate various natural life processes of plants. In addition, SA has an important role in plant signaling in response to pathogens and aphids. You should provide more correct definitions in the Introduction.
Reply:The article mentions the application of exogenous hormones, a chemical that functions similarly to endogenous hormones, so this description in the text should be accurate.
Comments 2:The second issue is the lack of information about the plant and the aphid species. The systematic position of the plant should be provided (e.g., botanical family, occurrence, ecological roles, volatiles attracting entomophages – what volatiles? – line 52). In the case of the aphid, more information would not hurt: for example, a short description of the life cycle, the host plant range to demonstrate that it is not a threat to cultivated plants, etc. The information on the aphid is necessary for the understanding of the results of the experiment: I assume that you studied the parthenogenetic population of the aphid. At this point in the life history, I assume that aphids reproduced parthenogenetically and viviparously. Therefore, the statement ‘eggs produced’ – line 220 – what eggs? Aphididae do not lay eggs when reproduce in the vegetative season on the secondary hosts. Literature provides the information that S. heraclei is a holocyclic host alternating species. In summer, it should be viviparous. Is it not? Please, clarify.
Reply:I have added descriptions of the family, genus, and ecological niche of Cnidium, as well as the names of its volatile compounds, and information that aphids do not harm the main crops.Aphids are indeed ovoviviparous, and the term "egg" was a mistake in my description.Preadult stage is the whole nymph stage
Comments 3:Table 1. Which data refer to what? Where is longevity?
Reply:The data in the 'total' column indicates that after the treatment, the lifespan of aphids has increased, making them more long-lived.Table 1 includes indicators such as the duration of the four nymphal instars, the duration of the adult stage, reproductive capacity, and the total lifespan.
Comments 4:Figure 1. What year of study does this figure refer to? Why only one year is presented whileFigure 2 presents the two-year results.
Reply: This data was collected in 2021 , due to the climate conditions in 2022, the number of aphids on the entire plot was very low that year, so data from 2022 was not included.However, the number of natural enemies is less affected by climate, so there are data for natural enemies for two years.
Comments 5:I listed only the most important issues crucial for the understanding of the results. As there are so many doubts, I can not evaluate the correctness of the interpretation. The article must be substantially improved. I am sure you have all necessary data but more work should be done. Moreover, the English of the text needs a lot of editing.
Reply:Thank you very much for your criticism and suggestions; I will make the necessary changes based on your feedback.

Round 2
Reviewer 1 Report
Comments and Suggestions for Authors
I could not find significant revisions in the comments, except for some error corrections. Please revise the manuscript more carefully, following the initial feedback.
Author Response
Comments 1:I could not find significant revisions in the comments, except for some error corrections. Please revise the manuscript more carefully, following the initial feedback..
Reply:I have revised the spelling, punctuation and grammar of the whole text. I put the details in the file. Please criticize and correct me

Reviewer 2 Report
Comments and Suggestions for Authors
Dear authors, thank you very much to consider my minor comments and most of majors ones. Here are my comments:
To my comment 11 you are asking me if it is possible to continue using ANOVA for this kind of data. My reply it is no, is not possible. ANOVA is inappropriate for time series data. You have all the data, you just need to run a different analysis like a GLMM with block as a random factor.
To my comment 18 you replied to me that since for other natural enemies there were no significant differences, they were not included. I suggest you to add a table with those natural enemies, it is important to know which natural enemies are not affected by salicyclic acid and indoleacetic acid.
Kind regards
Author Response
Comments 1:To my comment 11 you are asking me if it is possible to continue using ANOVA for this kind of data. My reply it is no, is not possible. ANOVA is inappropriate for time series data. You have all the data, you just need to run a different analysis like a GLMM with block as a random factor.
Reply:Thank you very much for your suggestions. We have re-analyzed the data, adopted GLMM method, and re-written it.
Comments 2:To my comment 18 you replied to me that since for other natural enemies there were no significant differences, they were not included. I suggest you to add a table with those natural enemies, it is important to know which natural enemies are not affected by salicyclic acid and indoleacetic acid.
Reply:I added additional species in the results section.

Reviewer 3 Report
Comments and Suggestions for Authors
I appreciated that you dealed with most of my suggestions, but I am not clear with your answers to the most complex points I raised :
- I think that you need a deeper analysis of the dynamics of both pests and predators using your data ; or if you can't, at least explain what your hypothesis would be and how you would test them
- definitions of peacks in graphs : when a dynamical growth starts before and with a way bigger level and lasting way longer, I can not see why you can tell that the second smaller one, that starts afterwards, is earlier .... please explain your analysis methodology
_ Furthermore as scientists, only stats shall tell us whether a phenomena is of significance or not ( sentences like "might be, may be, tendencies, hypothetically...." should be evoided), and please give the statistical tests results for every conclusion you have
Comments on the Quality of English Language
less flaws in language, but still needs revision
Author Response
Comments 1:I think that you need a deeper analysis of the dynamics of both pests and predators using your data ; or if you can't, at least explain what your hypothesis would be and how you would test them
Reply:The main idea of this paper is that plant growth regulators can provide more food for natural enemies by advancing the peak of aphids, thus increasing the number of natural enemies and improving the effectiveness of pest control. The current results also show that the number of natural enemies does increase after treatment. The effect of damage control needs more field experiments, but it cannot be realized for the time being due to conditions, and others may conduct subsequent studies.
Comments 2:definitions of peacks in graphs : when a dynamical growth starts before and with a way bigger level and lasting way longer, I can not see why you can tell that the second smaller one, that starts afterwards, is earlier .... please explain your analysis methodology
Reply:In this paper, the peak is defined as the period when the population number is the highest. I don't understand the first and second points of your suggestion, could you please explain them in detail.
Comments 3:Furthermore as scientists, only stats shall tell us whether a phenomena is of significance or not ( sentences like "might be, may be, tendencies, hypothetically...." should be evoided), and please give the statistical tests results for every conclusion you have
Reply:Thank you very much for your comments. We have shown conclusially that spraying plant growth regulators can advance the peak of aphids, increase their vitality and fertility, and increase the population of natural enemies.
However, due to the limited conditions, we did not carry out the verification of flower belt and crop intercropping. However, it has been proved in the literature that planting snake beds can increase the population number in the field, and aphids on the snake beds are important supplementary food for natural enemies. Therefore, this paper proves that exogenous hormones can increase the population number on the snake beds, so it is a reasonable speculation to improve their pest control ability. Of course, we have also done relevant experiments, but because the experimenter is not myself, we cannot add relevant data.

Reviewer 4 Report
Comments and Suggestions for Authors
Comment 1 Reply:The article mentions the application of exogenous hormones, a chemical that functions similarly to endogenous hormones, so this description in the text should be accurate.
My comment: Exactly, this is what I meant by improving the clarity. The expression 'exogenous hormones' is completely unclear. To improve the clarity, the expression ' exogenously applied hormones' should be used. 'Exogenous hormones' is the unnecessary mental shortcut.
For example, I suggest: 'Hormones for exogenous application' line 90. Please, make corrections in the text accordingly.
Comment 2. OK
Comment 3. The 'preadult' remained in the table. How do you define this stage here?
Comment 4. I accept the explanation, but you should include it in the text.
Comment 5. Language errors. You did not improve the text editing. Many errors remained and new errors appeared. For example: the names of the months should start with capital letters, in the description to the OY axis in Fig. 1A and 2A-D, it should be '... per..' not 'pear', in the explanation to Fig 1., there should be '5 plants' and not 5 plant. etc. Please, check the text very carefully or ask for professional help.
Author Response
Comments 1:I suggest: 'Hormones for exogenous application' line 90. Please, make corrections in the text accordingly.
Reply:Thanks for your suggestion, we have made the modification
Comments 2:The 'preadult' remained in the table. How do you define this stage here?
Reply:Preadult include four larva stages.It's the sum of these four periods
Comments 3:I accept the explanation, but you should include it in the text.
Reply:I added it under the caption
Comments 4:Language errors. You did not improve the text editing. Many errors remained and new errors appeared. For example: the names of the months should start with capital letters, in the description to the OY axis in Fig. 1A and 2A-D, it should be '... per..' not 'pear', in the explanation to Fig 1., there should be '5 plants' and not 5 plant. etc. Please, check the text very carefully or ask for professional help.
Reply:I have revised the spelling, punctuation and grammar of the whole text. I put the details in the file. Please criticize and correct me

Round 3
Reviewer 2 Report
Comments and Suggestions for Authors
Dear, authors! Here my comments:
1. Introduction: no comments
2. Materials and Methods:
-Sampling dates is not to be a fixed effect. A GLMM is a time series analysis, so you do not need to compare between dates.
-Line 299: you are comparing also Semiaphis heraclei. So you have two GLMM analysis: (1) Number of aphids with as fixed effects SA/IAA concentration, and block as random factor. (2) Number of natural enemies on C. monnieri with as fixed effects treatment (Water, SA and IAA) and year, and block as random factor. So you have only an interaction treatment*year and concentration*year
Results:
-Line 345: Those results from Anova are not correct. The degree of freedom is at least 4.
-Figure 1: C and D graphs are not necessary. In this part you just know the best SA or IAA concentration
-Line 355: In this section, you do not have to give statistical results for each data. Give the statistical results for year, treatment and the interaction. The post hoc multiple comparison test will say which treatment is different from the others for each natural enemy. Analysis of Deviance Table give you Chisq values, degree of freedom and P-value, it is what you need to mention here.
Author Response
-Sampling dates is not to be a fixed effect. A GLMM is a time series analysis, so you do not need to compare between dates.
-Line 299: you are comparing also Semiaphis heraclei. So you have two GLMM analysis: (1) Number of aphids with as fixed effects SA/IAA concentration, and block as random factor. (2) Number of natural enemies on C. monnieri with as fixed effects treatment (Water, SA and IAA) and year, and block as random factor. So you have only an interaction treatment*year and concentration*year
Reply:To assess the impact of various hormone treatments on the abundances of natural enemies and Semiaphis heraclei aphids, generalized linear mixed-effects models (GLMMs) were employed (Brooks et al. 2023). A zero-inflated Poisson model was used for natural enemies, incorporating fixed effects for treatments (water, 150 mg/L salicylic acid, and 150 mg/L indoleacetic acid), year (2021, 2022) and their second-order interactions, and a random effect for block. A zero-inflated negative binomial distribution was applied to aphid populations, incorporating fixed effects for different concentrations of SA and IAA and a random effect for block. The‘emmeans’package was utilized to carry out pairwise post hoc multiple comparisons of population numbers of natural enemies and aphids.
Line 345: Those results from Anova are not correct. The degree of freedom is at least 4.
Reply:Due to the updated analysis method, the description of the results has changed to: We observed the aphid abundance decreased significantly in the hormone treatments (df = 4,p = 1.184e-08) and 150mg/L SA treatment and IAA treatment were the most significantly decreased (p = 0.001 for 150mg/L SA treatment; p = 0.004 for 150mg/L IAA treatment).
-Figure 1: C and D graphs are not necessary. In this part you just know the best SA or IAA concentration
Reply:I think your suggestion is very pertinent, but this figure can also display the difference and maximum difference concentration more intuitively, so can we keep it?
-Line 355: In this section, you do not have to give statistical results for each data. Give the statistical results for year, treatment and the interaction. The post hoc multiple comparison test will say which treatment is different from the others for each natural enemy. Analysis of Deviance Table give you Chisq values, degree of freedom and P-value, it is what you need to mention here.
Reply:Replace part of the conclusion with the results of the analysis showed that for ladybugs, there were significant differences between years(Chisq=29.1673, df=1, p=6.639e-08), but no significant differences between treatments. However, O. minutus has significant differences between years(Chisq=15.4268, df=2, p=0.0004468) and treatment (Chisq=48.3202, df=1, p=3.62e-12). The results of pairwise comparison also showed that treatment would significantly increase the population of O. minutus in 2022(p=0.0004 for 150mg/L SA treatment and p=0.0004 for 150mg/L IAA).

Reviewer 3 Report
Comments and Suggestions for Authors
major point
I will try to reformulate my principal point : you can not conclude from your study that the aphid peack is advanced with the use of the hormone, as the aphid population start to grow in the control way sooner and higher (I would rather even say that the hormones delays the aphid population growth, and prevent it for growing further !), and the population level reached at the IAAS peacks is reached way sooner in the control : all that i can see here, and for this i agree with you, is that this product decrease the aphid population. If the control peack was distributed in a narrower period arround its climax I would understand. If you think that the difference between lab and field is the predators, you definitivelly should consider make some tests in planta in cages to control this and complete your study.
Moreover, you can not conclude that it would be a way to attract predators sooner, as the aphid level is way way higher sooner in the control, therefore way more attractive potentially, and you did not observe significative differences in the arrival dates of any of the predators in your test.
l479 : i would rather say "we observed more O. minutus and P. japonica in IAA treated plants " as you did not test anything on them directly, so it could be a direct or an indirect effect
l483 -484 you can not conclude that
l518 : Why would natural enemies impact more aphid in treated plants and not be significantly more present ? do you think that the treatment make the prey more "preyable" ? ease the access, or prevent the preys to flee or something like this ?
minor edits
l81 : improve instead of Improve
l84 : replace "however natural enemies ... pests" by just "therefore", as you already added "advencing the entry tim ...."
l86: "an"insect-attracting...
l 223 : C. monnieri in italic
l 233 : what was the origin of the C. monnieri seeds you use ? wild plants ? or give the brand
l239 : five in letters
l246 : what material did you use to spray your crops
l477 "and" instead of "but"
l490 : in the present study
l504 : . instead of ,
Comments on the Quality of English Language
still needs to be improved. Besides the spelling, you would gain in fluidity by having it revised by a native , as sentences are sometimes hard to comprehend. And you have to do this during revision so we can evaluate the finished work
Author Response
I will try to reformulate my principal point : you can not conclude from your study that the aphid peack is advanced with the use of the hormone, as the aphid population start to grow in the control way sooner and higher (I would rather even say that the hormones delays the aphid population growth, and prevent it for growing further !), and the population level reached at the IAAS peacks is reached way sooner in the control : all that i can see here, and for this i agree with you, is that this product decrease the aphid population. If the control peack was distributed in a narrower period arround its climax I would understand. If you think that the difference between lab and field is the predators, you definitivelly should consider make some tests in planta in cages to control this and complete your study.
Moreover, you can not conclude that it would be a way to attract predators sooner, as the aphid level is way way higher sooner in the control, therefore way more attractive potentially, and you did not observe significative differences in the arrival dates of any of the predators in your test.
Reply:First of all, my definition of peak population should be the period when the population is at its maximum, so I don't see a problem with my description. Second, we found that the treatment increased the aphid population in the laboratory but decreased the population in the field, and combined with the population of natural predators, we concluded that it may be because more aphids attracted predators, and the predators preyed on the aphid population. Finally, I think the cage experiment you said is quite correct, but I am no longer in this research group, so it is difficult to complete.
l479 : i would rather say "we observed more O. minutus and P. japonica in IAA treated plants " as you did not test anything on them directly, so it could be a direct or an indirect effect
Reply:We came to this conclusion using population data from field surveys
l483 -484 you can not conclude that
l518 : Why would natural enemies impact more aphid in treated plants and not be significantly more present ? do you think that the treatment make the prey more "preyable" ? ease the access, or prevent the preys to flee or something like this ?
Reply:Predators prey on aphids, so they reduce their population rather than increase it. Snake bed is a plant in the umbelliferaceae family. It may be that treatment makes it easier for predators to prey on aphids or makes plants release more volatiles to attract predators. There may be many reasons for this, which still need further research.
improve instead of Improve
Reply:improve has been change to Improve
replace "however natural enemies ... pests" by just "therefore", as you already added "advencing the entry tim ...."
Reply:"however natural enemies ... pests" has been change to "therefore"
l86: "an"insect-attracting...
Reply:a has been change to an
- monnieri in italic
Reply:C. monnieri has been change to C. monnieri
what was the origin of the C. monnieri seeds you use ? wild plants ? or give the brand
Reply:The seeds were collected after maturity from plants previously grown in fields in Shandong Province by the laboratory. I had added this in material
five in letters
Reply:Corrections have been made in the text
what material did you use to spray your crops
Reply:We prepared a 10L garden watering can and prepared the solution 1h before each spray. I had added this in material
"and" instead of "but"
Reply:"and" has been change to "and"
in the present study
Reply:Corrections have been made in the text
. instead of ,
Reply:. has been change to ,

Round 4
Reviewer 2 Report
Comments and Suggestions for Authors
Dear authors, many thanks for the new version. Here are a few comments.
-line 208: correct the size of all the natural enemies, it looks bigger than the rest.
-review all the scientific names in titles.
Author Response
--line 208: correct the size of all the natural enemies, it looks bigger than the rest.
Reply:Thanks for your suggestion, I have checked this part and confirmed that there is no problem.
--review all the scientific names in titles.
Reply:Thanks for your suggestion, I have checked the Latin scientific name of the full text and made sure that the font and scientific name are correct.

Reviewer 3 Report
Comments and Suggestions for Authors
thank you for adding more stats.
You should add the necessity to do cage experiments in the perspective or discussion to validate your conclusion. Furthermore, I liked your answer to my question on preys availability (l518 previous review), you should also add something about it in your discussion.
l167 point missing
l120 two points
l113 "sity .... winter" to be suppressed
l337, l347 extra spaces
ref 28 (l461), format needs to be corrected
Comments on the Quality of English Language
needs revision
Author Response
You should add the necessity to do cage experiments in the perspective or discussion to validate your conclusion. Furthermore, I liked your answer to my question on preys availability (l518 previous review), you should also add something about it in your discussion.
Reply:Thank you for your advice. I added a discussion on the possibility of predation in the discussion section and a suggestion to add cage experiments in subsequent experiments to make the article more accurate.
l167 point missing
Reply:I have added this point
two points
Reply:I have taken off the extra dots
"sity .... winter" to be suppressed
Reply:I have suppressed "sity .... winter"
extra spaces
Reply:I have taken off extra spaces
format needs to be corrected
Reply:Thanks for your suggestion, I have modified the format of the document
